# Ultraweak Photon Emissions as a Non-Invasive, Early-Malignancy Detection Tool: An In Vitro and In Vivo Study

**DOI:** 10.3390/cancers12041001

**Published:** 2020-04-18

**Authors:** Nirosha J. Murugan, Michael A. Persinger, Lukasz M. Karbowski, Blake T. Dotta

**Affiliations:** 1Behavioural Neuroscience & Biomolecular Science, Laurentian University, 935 Ramsey Lake Road, Sudbury, ON P3E 2C6, Canada; Nirosha.Murugan@tufts.edu (N.J.M.); mpersinger@laurentian.ca (M.A.P.); oligma@gmail.com (L.M.K.); 2Department of Biology, Algoma University, Sault Ste. Marie, ON P6A 2G4, Canada; 3Department of Biology, Laurentian University, Sudbury, ON P3E 2C6, Canada

**Keywords:** ultraweak photon emission, cancer diagnostics, pancreatic cell cultures, malignancy, mice developing tumors, discriminant functions

## Abstract

Early detection of cancer improves treatment options and increases survival. Building upon previous demonstrations that ultraweak photon emissions (UPE) could be measured to detect cancers, we designed an early detection protocol to test malignancy in both in vitro and in vivo systems. Photons were measured for 100 s from plates containing ~1 million malignant or non-malignant cells from 13 different types of human and mouse cell lines. Tumor cells displayed increased photon emissions compared to non-malignant cells. Examining the standardized Spectral Power Density (SPD) configurations for flux densities between 0.1 and 25 Hz (Δ*f* = 0.01 Hz) yielded 90% discriminant accuracy. The emission profiles of mice that had been injected with melanoma cells could be differentiated from a non-malignant reference groups as early as 24 h post-injection. The peak SPD associated with photon emissions was ~20 Hz for both malignant cell cultures and mice with growing tumors. These results extend the original suggestion by Takeda and his colleagues (2004) published in this journal concerning the potential diagnostic value of UPEs for assessing proliferations of carcinoma cells. The specificity of the spectral profile in the 20 Hz range may be relevant to the consistent efficacy reported by several authors that weak magnetic field pulsations within this frequency range can diminish the growth of malignant cells in culture and tumor weights in mice.

## 1. Introduction

The observation made almost a century ago [1] that all living systems emit discrete numbers of ultraweak photons as well as the reiteration of this phenomenon by Popp and his colleagues [2] suggested that a non-invasive method might someday be used to discern the states of cells within the human body. Their physical bases and role within biological systems has remained a relevant topic of investigation [3,4,5,6]. Takeda and colleagues have reported that biophoton emissions—also referred to as ultraweak photon emissions (UPE)—could be measured to assess the proliferation of carcinoma cells [7]. Since then, several independent experiments have supported their original observations [8,9]. Although raw photon emissions or power flux density can be revealing, spectral power densities (SPD) of photon emissions reveal the underlying complexity of the distribution of the power over time.

From both theoretical and practical perspectives, the measurement of photons as potential early diagnostic indicators of cell anomalies within the body is advantageous. Photons are the bases, or at least the strong correlate of, changes of energies across electron orbits in the atomic and molecular reactions that drive living systems. Changes in numbers of photon emissions have been measured as a function of chemical reactions related to integrity of the plasma membrane and mitochondria activity [10,11]. Dotta and colleagues showed that the power flux densities of photon emissions from melanoma cells were within the range of 10 ^−11^ to 10 ^−10^ W·m ^−2^ and would be equivalent to about 10 ^−20^ J per cell membrane per 2–3 s [12]. This value is the energy from the electric forces between potassium ions that contribute to the resting membrane potential and to many forms of ligand sequestration [13]. Trushin [14] and Fels [15] have argued that photon emissions are not simply artifacts of cell metabolism but could mediate information between cells. The procedure for measuring UPEs is not intrusive and requires relatively inexpensive digital photomultiplier detector units arranged with appropriate geometries. Since the goal is not to image but rather to detect cell states, the major challenge has been to discern the characteristics of the photon missions in a reasonable (brief) time.

Contemporary models that explain the development of cancer or the proliferation of malignant cells describe discrete molecular signaling pathways. These are powerful and predictive models. For photon measurements to have any utility to compliment what is offered by molecular biology, specific characteristics of photon measurements should be relatable in a quantitative manner to specific components of molecular pathways in cancer cells. Demonstrating the basic principle, Dotta and colleagues [16] tested Cosic’s model [17,18] for predicting the peak wavelength emissions based upon the amino acid sequence in proteins and measured specific narrow bands of photon emissions from melanoma cells stressed by remaining at room temperature. Application of specific filters that allowed passage of photons within the visible range within a ~10 nm increment wavelength predicted by Cosic’s Molecular Resonance Recognition model showed that facilitators or inhibitors of the target reaction either increased or decreased the photon emissions.

The primary limit of wide-band (visible range) photomultiplier tubes (PMTs) is that specific wavelengths associated with different pathways cannot be easily differentiated. Dependence upon the number of photons per unit time or radiant flux density can be influenced by the sensitivity range of the PMT and the position of the sensor in space. Spectral Power Density obtained from Fast Fourier Analyses is an advantageous data transform, when standardized and applied optimally, to discern the information within the amplitude fluctuations of the photon emissions over the sampling period. It is a method of detection rather than of imaging. For example, SPD reveals shifts in the distribution of photon emission power for malignant cells [19] and preparations of microtubules [20] from application of experimentally designed resonant weak magnetic fields even though there was no significant shift in the total number of photons emitted. Redistributions of energy within a closed system is a central feature of thermodynamics. This robust observation was the basis for the following experiments that reiterated the marked discriminating capacity of SPD profiles for differentiating normal and malignant cells lines as well as mice with and without cancer. In this paper, we report that specific increments of temporal variations of SPD obtained for very brief (100 s) samples of photons measured by wide spectrum digital photomultiplier units can be employed to discriminate between malignant and non-malignant conditions.

## 2. Materials and Methods

### 2.1. Cell Lines

A total of 13 cell lines involving human and animal malignant and non-malignant origins were tested. The cell lines were: AsPC-1, Capan-1, CFPAC-1, B16-BL6, BxPC3, HBL 100, HEK 293, HELA, HPAF-11, HSG, Hs 578T, MCF7, and MDA MB 231. Cells were kept in 150 × 20 mm tissue culture plates (Sarstedt, Laval, QC, Canada) in Dulbecco’s Modified Essential Medium (DMEM, Hyclone, Logan, UT, USA supplemented with 10% fetal bovine serum, 100 ug/m streptomycin, and 100 U/mL penicillin (Invitrogen, Burlington, ON, USA). Cells were sub-cultured 1:5 every 3–4 days and incubated at 37 °C in 5% CO_2_, 95% humidity.

### 2.2. Animal Procedures

In a series of 4 different experiments, the whole-body photon emissions from a total of 47, C57 male mice were tested intermittently over about 20 days. The mice had been acclimatized to the colony conditions and housed 4 per cage for at least one month before beginning the experiments. The light-dark cycle was 12:12 h with the offset of lights at 20 h local time and the room temperature was controlled around 21 °C ± 1 °C.

Half of the mice were injected subcutaneously within the right flank with a 100 μL Hamilton syringe with 5 million B16-BL6 cells that had been sub-cultured the same day and extracted in 10 cc of media [21]. The other half of the mice served as the reference groups. Mice were divided into three groups, where the first group was injected with B16-BL6 cells that had been exposed to UV for 60 min, which killed the cells (confirmed by live/dead staining). The second group was injected with living B16-BL6 cells, and the third group received no injections. The purpose was to discern if simply injecting cells produced an alteration in photon emissions. The experimental groupings can also be seen in Figure 1. The number of mice in each group was: control condition *n* = 16, tumor condition *n* = 23, and UV irradiated *n* = 8.

The whole-body photon measures were completed on 4 different days over the course of 21 days of tumor development. The specific days differed for each experiment. However, measurements were only performed on day 1, 7, 10, or 13 post-injection. This was completed to ensure there were no unusual windows that would have been missed with a narrower observation period. When all experiments were combined, the mice that received the subcutaneous injections and their controls were measured on post-injection days 1, 7, 10, and 13.

### 2.3. Photomultiplier Measurements

For the cells, the photon emissions for a total of 6 replicates for each of the 14 cell lines were measured by two different digital PMTs (DM0089C; DM0090C). Although their sensitivity overlapped, the 89C extended more into the UV range, which increased the spectral window within which UPEs were detected and, thus, increasing background UPE detection (i.e., dark counts). Their average dark counts were 2500 and 15 photons per second, respectively. The plate whose cells had been split 3–4 days previously was placed over the aperture of the PMT. It was contained within a black painted box covered with several layers of thick black terry cloth in a dark room. The numbers of photons per 20 ms (50 Hz sampling) were recorded for 100 s by a laptop that was external to the PMT. This sampling rate reflected the upper limit of the software (SENS Tech Counter/Timer software, Version 2.8, Build: 11).

For animal measurements, each mouse was anesthetized using 50 mg/kg Ketamine intra-venousinjections and placed in a wooden box (6 cm by 10 cm by 6 cm) with a transparent lid placed on top of the box to allow detection with the PMT. This equipment containing the mouse was then placed into a second black box (30 cm by 30 cm by 30 cm) and covered with several layers of black terry cloth towels. The numbers of whole-body photons were recorded for two successive 100-s increments on each day of measurement. To ensure that placing mice into the same box was not contributing to the artifact, two of the experiments involved a single box for each mouse that was only employed for that mouse over the measurement period. After each measurement, animals were euthanized by asphyxiation (CO_2_) and internal cervical dislocation.

### 2.4. Data Analysis

All statistical analyses involved SPSS 16 software for PCs. For an in vitro study, the mean numbers of photon emissions during the 100-s sample were obtained for each replicate for each cell line. The means and standard deviations for the numbers of photons per s were calculated. Since there were two PMTs with different dark counts and sensitivities, measurements for the cell lines completed with the same PMT were compared. For an in vivo study, the mean numbers of photon emissions per s for the first 100-s samples for each day were calculated. A second 100-s sample was also recorded.

Spectral analyses were completed for each trial for all cell lines. Since the sampling rate was 50 Hz, the Nyquist Limit was 25 Hz for the amplitude variations of the spectral power. The frequency increment for this sampling rate for this duration was 0.01 Hz. A step-wise linear discriminant analysis was completed for the spectral profiles. Two types of analyses were performed. The first was malignant-nonmalignant cell comparisons, which included the human breast cell and pancreatic cell lines. The second analysis was performed to discern the strength of the function (i.e., the canonical correlation) for discriminating between non-malignant and malignant cells.

The spectral analyses for the photon emissions from the mice with or without growing tumors were completed in a similar manner. This means that the range of the power variations of the photon output was between 0 and 25 Hz. The means and standard deviations for the standardized power outputs for each condition (tumor vs. reference) were calculated for each day of measurement. Separation of the dispersion indicators indicated statistically significant differences.

## 3. Results

### 3.1. Cell Line Photon Emission Profiles

The means and standard deviations (SD) for the numbers of photons per second per cell line during the 100-s sample within a few minutes after being removed from incubation are shown in Table 1 and Table 2. The two tables reflect the results of two different PMTs with slightly different sensitivities within the visible wavelengths. Figure 2 [F (1,138) = 3.7, *p* = 0.05, eta^2^ = 0.026] shows the means and standard error of the mean (SEM)for malignant and non-malignant cells for human pancreas and breast origins. Since they were measured by the same PMT, the photon counts could be directly compared. The malignant cells emitted more photons than their non-malignant reference cells.

The results of the spectral analyses were even more revealing. Since the photon numbers were standardized within each sample, the spectral profiles could be compared across the instrumentation. As shown in Figure 3, the SPD profiles for the malignant pancreatic cells were significantly different than for the non-malignant counterparts (*p* < 0.001). The malignant cells displayed less power when compared to reference cells at 0.2 Hz, 12 Hz, and 23.8 Hz, and more power when compared to reference cells at 7.6 Hz and 19.7 Hz. Discriminant analyses for the malignant-normal cell pairs from a given tissue indicated that four to five specific spectral frequency increments differentiated the two populations with 83% accuracy.

On the other hand, when all of the cells were divided into malignant vs. non-malignant groups and all of the increments of frequency from the SPD were allowed to enter the analyses for the optimal function, the first five variables (Δf) discriminated only about 70% of the malignant vs. non-malignant cases. The only SPD increment that was significant different between malignant and non-malignant cells was 22 Hz.

### 3.2. Mice Whole Body Photon Emission Profiles

The most conspicuous change in photon emissions involved the presence or absence of the mouse in the box. An example of the means and standard deviations for the numbers of photons per s when the mouse was present or not in the box is shown in Figure 4 [F(1,53) = 19.65, *p* ≤ 0.001, eta^2^ = 0.27]. The means and standard error of the mean for the numbers of photons per s during the successive 100-s sampling of mice with and without injections of melanoma cells (viable or killed by UV). Therefore, developing tumors are shown in Figure 5 [F(8,137) = 11.6, *p* ≤ 0.001, eta^2^ = 0.40].

Compared to controls (non-injected mice) in which the mean raw photon counts did not change significantly 24 h, 7 days, or 13 days after the experiment began, the group that had been injected with tumor cells (alive or dead) displayed about 300 photons more per s within 24 h. One week later, since the photon emissions from the controls were similar to the beginning of the experiment, both tumor groups displayed fewer (about 300 photons) emissions. At approximately two weeks post injection, both groups’ photon emissions had normalized as indicated by the similarity to that of the control group.

Again, the results of the spectral analyses were quite revealing. There was a conspicuous difference between mice with developing tumors and those without 24 h post injection. Where mice with developing tumors had significantly less SPD within the 22.7 Hz when compared to control mice, and mice injected with dead cells [18 (6.7) vs. 23(6.7) vs. 25.7(7.2) respectively, Means and SD, *p* = 0.033, eta^2^ = 0.15]. Additionally, an SPD effect was observed from mice whose photon emissions were measured 19 days post injection (*n* = 6, Control = 3, Tumor = 3). Results indicated that mice with fully developed tumors (19 days post injection) displayed higher SPD scores when compared to non-injected controls within the 22.6 Hz band [28(4.4) vs. 17.4(4.4) respectively, Means and SD, *p* = 0.042, eta^2^ = 0.68].

## 4. Discussion

Takeda et al.’s [7] classic report describing the efficacy of UPEs as indicators of carcinoma cell proliferation extended the concepts developed by Gurwitch [1], Popp [2], and van Wijk [21]. If the numbers of photons are coupled to the rate or number of biochemical reactions involved with cell division, then one would expect a systematic increase in photon flux density in organisms with proliferation. From this perspective, the question as to whether or not the photons are a source of information exchange between cells [15] or artifacts of metabolism [12] is not critical. From an empirical and clinical perspective, the enhanced flux density can be employed as a differential diagnostic.

An extension of Helmholtz’ principle that energy changes “in form” rather than being diminished or increased is that the power (Joules per second) may remain more or less the same. However, the expression, i.e., the temporal pattern of increments of amplitude, may change. Takeda [7] noted that the significant peak in photon wavelength was about 530 nm. Their sampling time was 93 h. We measured shifts in the peak photon wavelengths for melanoma cells that had been removed from incubation and remained at room temperature over a course of about 10 h [16]. Although there was an attenuation of power density within about an hour, what was more revealing was the shift in peak wavelength from near-infrared to near-ultraviolet over several hours. The latency between beginning and end of the shift coincided with the mitotic cycle.

Dotta [22] extended this concept of “temporal shifting” of spectral power densities to photon emissions from microtubule preparations exposed to weak magnetic fields that simulated the patterns associated with components of long-term potentiation. We found that the photon power density did not change significantly, even though the shift in the spectrum density for that power was very clear. Spectral power density has been employed in many other disciplines where shifts in energy magnitude would not be affected but the mixtures of frequencies or phases would. As an analogy, the power output of a radio generating static or a broadcast message is the same. The difference is the pattern of expression.

In the present study, SPDs of only 100 s of photon measurements were powerful discriminators of malignant vs. non-malignant cells from the same tissue. Each different type of cell revealed different SPD patterns. Such fine distinctions should add to the utility of the method for ultimately differentiating different types of tumors during whole body measurements. The accuracy of discrimination diminished when we attempted to differentiate the SPD profiles and all malignant cells from different cell lines from all non-malignant cells. This is expected. Different types of malignant cells would exhibit different profiles that would add much greater variability in the denominator of the statistical operator and obscure simple dichotomous differentiation.

What emerged was the relatively reliable observation that the average of all malignant cells differed from normal cells with photon emission power frequencies around 22 Hz. Whether or not this is a general signature of malignant cells must still be discerned. We did measure a similar peak in the mice that had been injected with melanoma cells. However, they did not exhibit this pattern immediately but only during the latter stages of tumor development when the tumor was largest and when most mice began to lose body weight. The profile was different than 24 h after injection of the cells.

The conspicuous difference in photon numbers when mice were in or not in the box was about 500 photons per second. Although marginally significant, the net increase in photon emissions over time for mice with developing tumors was about 10% greater than the non-injected controls. The tumor weights at that time were between 1 and 2 g. Hence, even quantitatively, the meta-analytic average of the mean photon numbers per second over time for the tumor-containing mice compared to the reference group would be consistent with the additional mass of the tumor.

Whereas UPEs were initially greater in numbers within groups with tumours (with or without UV-treatment) relative to un-injected controls, changes over time were apparent. In particular, the tumour group displayed a sustained decrease of UPEs on day 7 and day 13 relative to initial values. The UV-treated group displayed a decrease of UPEs on day 7, which was followed by an increase on day 13. While it is not fully clear why long-term UPEs might shift as a function of the different conditions, it is clear that un-injected controls remain reliably stable. We hypothesize that different UPEs over time may be reflective of sequential biological processes such as rapid proliferation, growth, and even cell death.

The SPD peaks around 20 and 22 Hz for differentiating tumor cells in culture or mice with and without tumors may not be a coincidence. This peak refers to power of the photon output rather than the frequency of the photons’ wavelength, as measured by Takeda [3] or Dotta et al. [16,20]. It may be relevant that several authors who have applied mT to uT magnetic fields ranging from 20 to 50 Hz to breast cancer cells [23], 8 to 20 Hz to melanoma cells where T-type calcium channels were blocked or facilitated [24], or 8 to 20 Hz for mice with growing melanoma tumors [21], noted the importance of this same band of power fluctuation. The potential therapeutic relevance of combining the SPD configuration for photons with the frequency of these applied fields remains to be investigated.

## Figures and Tables

**Figure 1 cancers-12-01001-f001:**
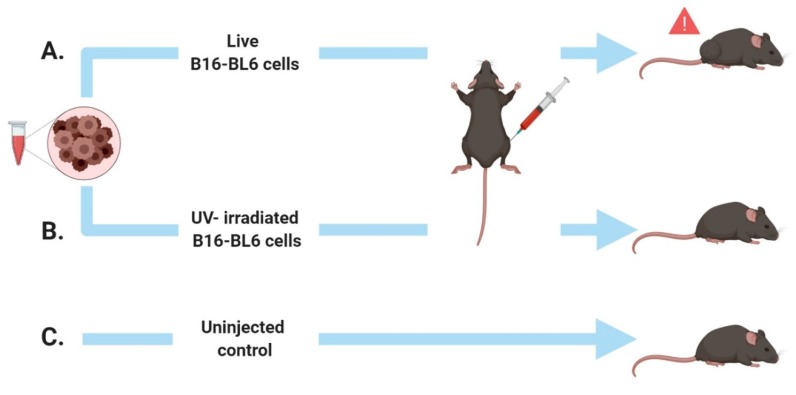
Design schematic for in vivo experiments. Male C57 mice received a right flank injection of B16-BL6 cells, expressing large palpable tumors 19 days post-injection (**A**). Other mice received negative control injections of UV-irradiated B16-BL6 cells (**B**) or no injections, (**C**) which did not induce tumor growth.

**Figure 2 cancers-12-01001-f002:**
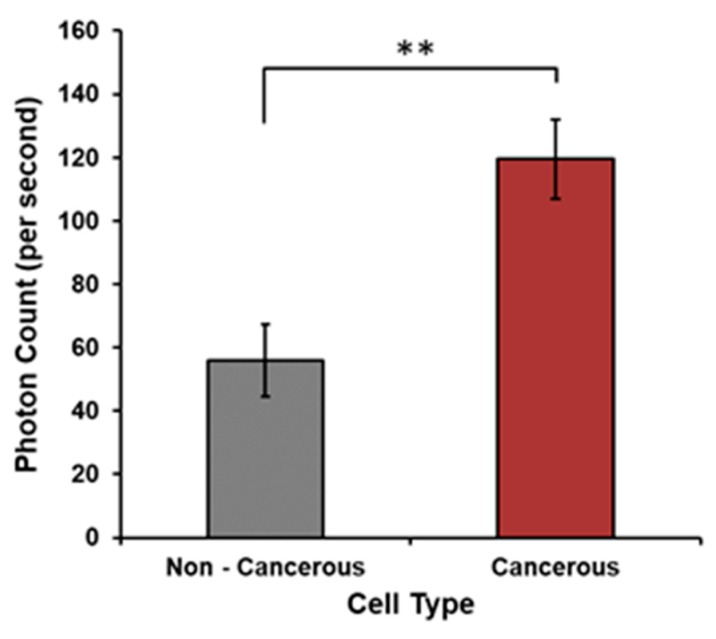
Photon counts per second from malignant and non-malignant cells show a significant decrease in photon counts (** = *p* < 0.05) in non-cancerous cells as compared to the cancerous phenotypes. Means and SEMs presented.

**Figure 3 cancers-12-01001-f003:**
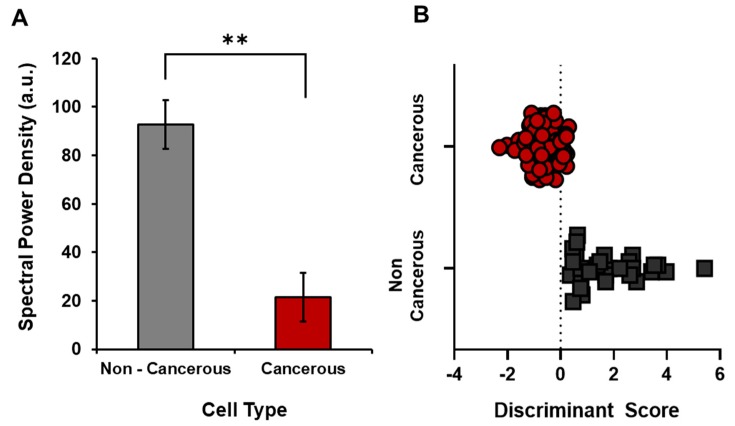
(**A**) Spectral power densities for 0.2 Hz frequency band can accurately classify 83% of 113 cases for cancerous and non-cancerous pancreatic cell lines (** *p* < 0.001). (**B**) Discriminant score for cancerous and non-cancerous cell lines. Scores were computed with the following discriminant function (=0.2 Hz Freq * (0.023–1.023).

**Figure 4 cancers-12-01001-f004:**
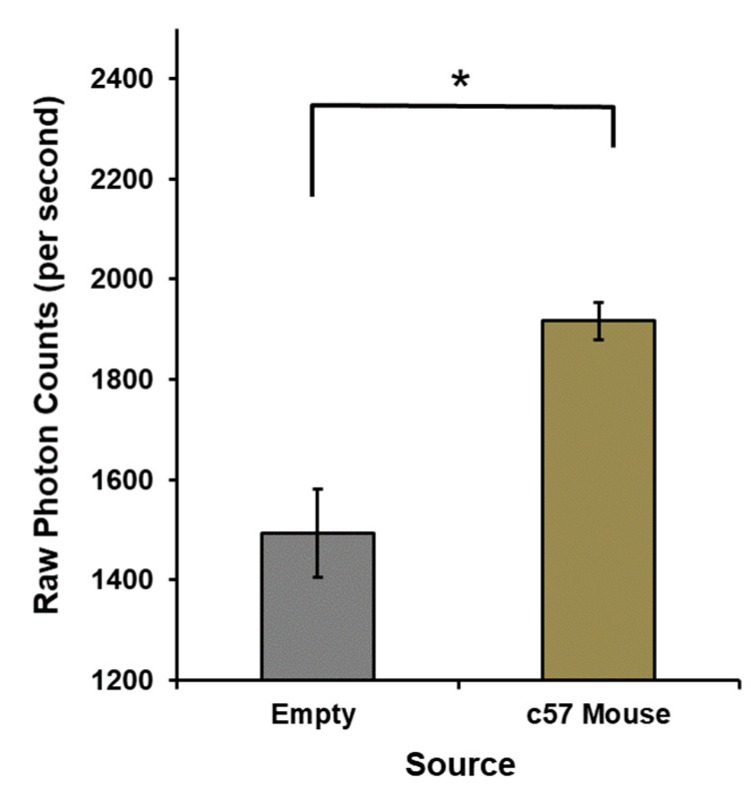
Raw photon counts per second within the dark-adapted box when no mice were present and when c57 mice were present. Means and standard error or the means are presented (* = *p* < 0.05).

**Figure 5 cancers-12-01001-f005:**
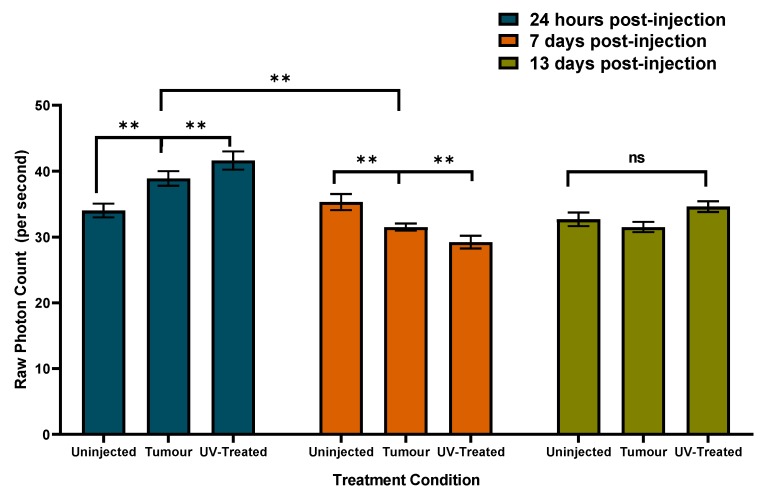
Photon counts per second by injection group (control = no injection, Tumour = melanoma cells injected, UV = ultraviolet irradiated melanoma cells injected) and time from injection day. Means and SEMs presented. (** = *p* < 0.05, ns = not significantly different).

**Table 1 cancers-12-01001-t001:** Photons per second from seven different cell types as measured from photomultiplier tubes (PMT 1) (higher dark counts). Mean and standard deviations are presented.

Cell State	Cell Type	Mean (SD)
**Healthy**	HSG	2631 (±828)
	HEK 293	2518 (±382)
	HBL 100	3070 (±662)
**Cancerous**	MCF-7	3138 (±6897)
	MDA MB 231	3240 (±943)
	B16 BL6	2927 (±1095)
	HeLa	2696 (±786)

**Table 2 cancers-12-01001-t002:** Photons per second from 11 different cell types as measured from PMT 2 (lower dark counts). Means and standard deviations are presented.

Cell State	Cell Type	Mean (SD)
**Healthy**	HS5 78T	79 (±25)
	HEK 293	81 (±49.5)
	HBL 100	80 (±56)
**Cancerous**	MCF-7	139 (±169)
	MDA MB 231	76 (±32)
	B16 BL6	229 (±175)
	HPAF-11	62 (±31)
	AsPC-1	11 (±71)
	Capan-1	116 (±83)
	BXPC3	128 (±119)
	CFPAC-1	120 (±90)

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
