# Peer review of "Ultraweak Photon Emissions as a Non-Invasive, Early-Malignancy Detection Tool: An In Vitro and In Vivo Study"

_cancers, 2020, doi:10.3390/cancers12041001_

Round 1

Reviewer 1 Report

Major concerns:

  1. The authors have used expressions such as ultra-weak photon emission, biophotons etc., I recommend that authors introduce in the section “Introduction”, the general terms used to identify the same phenomenon.
  2. In Figure 4, raw photon counts per second within the dark-adapted box when no mice were present and in the presence was measured. I cannot find information on whether mice were on under anaesthesia?
  3. What is the reason for lower photon counts in the 7-day post (tumour group)? The authors should comment and discuss the point in details.

Minor points:

Line 19- Incomplete sentence “Photons per?”

Line 33- “Ultraweak” formatting error

Line 54- Authors are advised to consider adding the following citations

  1. E. Lozneanu, M. Sanduloviciu, Physical basis of biophoton emission and intercellular communication, Rom. Rep. Phys. 60 (3) (2008) 885-898.
  2. V.P. Kaznacheev, L.P. Mikhailova, N.B. Kartashov, Distant intercellular electomagnetic interaction between two tissue cultures, Exp. Biol. 89 (3) (1980): 337-339.
  3. C. Rossi, A. Foletti, A. Magnani, S. Lamponi, New perspectives in cell communication: Bioelectromagnetic interactions, Semin. Cancer Biol. 21 (3) (2011) 207-214.
  4. A. Prasad, C. Rossi, S. Lamponi, P. Pospíšil, A. Foletti. New perspective in cell communication: potential role of ultra-weak photon emission. Joornal of photochem. Photobiol B: Biology 139 (2014) 47-53.

Author Response

Thank you for your helpful comments and suggestions. We have provided a point by point review to your comments. 

Reviewer #1
We would like to thank the reviewer for their helpful comments which have significantly improved the quality of the manuscript. The revised manuscript addresses each of the reviewer’s concerns.

Major Concerns:
1) Comment: The authors have used expressions such as ultra-weak photon emission, biophotons etc., I recommend that authors introduce in the section “Introduction”, the general terms used to identify the same phenomenon.
1) Response: We agree with the reviewer that the nomenclature should be consistent to increase clarity and avoid confusion. On Page 1, Line 40-41, we have added a statement which explains that the terms ultraweak photon emissions and biophotons are synonymous within the context of the manuscript. Further, we have used the consistent abbreviation UPE throughout the manuscript.

2) Comment: In Figure 4, raw photon counts per second within the dark-adapted box when no mice were present and in the presence was measured. I cannot find information on whether mice were on under anaesthesia?
2) Response: To clarify, we have included more detail in the revised manuscript under the section heading 2.3. Photomultiplier Measurements to further describe the condition of the mice before, during, and after the PMT measurement procedure including anesthetization and the method of euthanization.

3) Comment: What is the reason for lower photon counts in the 7-day post (tumour group)? The authors should comment and discuss the point in details.
3) Response: Both tumour subgroups (Tumour and UV-Treated) varied as a function of time. In the case of the tumour group, photon counts drop on day 7 and remain low on day 13.
UV-Treated samples displayed decreased photon counts on day 7, followed by an increase on day 13. Though it is not fully known why this is, we have provided a brief discussion (1 paragraph) in the revised manuscript’s discussion section on Page 9.

Minor Concerns:
1) Comment: Line 19 – Incomplete sentence “Photons per?”.
1) Response: We have omitted the word “per”, which completes the sentence.

2) Comment: Line 33 – “Ultraweak” formatting error.
2) Response: The bolded word has been changed to the correct format.

3) Comment: Line 54 – Authors are advised to consider adding the following citations:
1.E. Lozneanu, M. Sanduloviciu, Physical basis of biophoton emission and intercellular communication, Rom. Rep. Phys. 60 (3) (2008) 885-898.
2.V.P. Kaznacheev, L.P. Mikhailova, N.B. Kartashov, Distant intercellular electomagnetic interaction between two tissue cultures, Exp. Biol. 89 (3) (1980): 337-339.
3.C. Rossi, A. Foletti, A. Magnani, S. Lamponi, New perspectives in cell communication: Bioelectromagnetic interactions, Semin. Cancer Biol. 21 (3) (2011) 207-214.
4.A. Prasad, C. Rossi, S. Lamponi, P. Pospíšil, A. Foletti. New perspective in cell communication: potential role of ultra-weak photon emission. Joornal of photochem. Photobiol B: Biology 139 (2014) 47-53.

3) Response: The provided references are excellent; we have added them to the revised manuscript.

Reviewer 2 Report

Murugan et al. present an appealing demonstration for using ultra-weak photon emission as a means to distinguish cancer and non-cancer samples. The authors report a higher photon emission from the cancerous samples. The discussion and the overall experiment is well designed and enjoyable to read.

However, I find a severe lack in the convincing of the results in the current form of the MS.

Line 126: The PMTs used- DM0089C (bialkali, 28%, darkcount~ 50-200 photons/sec); DM0090C (S20, 18% QE, dark count~ 1000 photons/sec) demonstrated 2500 and 15 photons/sec dark counts respectively. Why this discrepancy? Was the discriminator threshold set arbitrarily?

The variation from the PMTs (low dark counts, low QE, UV-VIS) or (high dark counts, high QE, VIS) in Tables 1 and 2 is not entirely intuitive.
Table 1 and Table 2 are the same except for the last four rows!

Labels in Figures 2 and 3 are not consistent. Use"cancer" or "malignancy".
The x-axis for Figure3 is missing.
Figure 3 bar plot doesn't give any additional information than figure 2, without details on the density estimation.
Fig5: Consider using color-coding or similar to help read the graph easier.

50Hz sampling: is this limited by the amplifier or the USB bandwidth? This seems too low sampling for most biomedical imaging units (1-200 kHz). Was this PMT unit a cost-efficient implementation?

How was the discriminant analysis performed? No details on the method are presented. Why are the spectral graphs/power density graphs not presented?

If authors are willing, I suggest pushing the time-series data to open-analysis/ open-data platforms for community efforts. Also, additional work to identify the significance of these count rate differences, apart from a regular bar plot, would add some rigor to the analysis.

Line19: "photons per" -> it is not clear what authors mean here (area, time or others)

If these comments are addressed, I find the article interesting for cancer diagnostics and the bio-medical community. 

Author Response

We have provided a point by point response to you helpful comments and suggestions. Thanks you. 

Reviewer #2
We would like to thank the reviewer for both their compliments and suggestions regarding the manuscript. We appreciate the thorough reading and agree that several major changes were needed to increase the quality of the paper – particularly in reporting results. The revised manuscript includes these changes that we hope are to the satisfaction of the reviewer.

1) Comment: The PMTs used- DM0089C (bialkali, 28%, darkcount~ 50-200 photons/sec); DM0090C (S20, 18% QE, dark count~ 1000 photons/sec) demonstrated 2500 and 15 photons/sec dark counts respectively. Why this discrepancy? Was the discriminator threshold set arbitrarily?
1) Response: We agree that the use of two different and independent instruments can be confusing; however, the convergence of results indicates that the observed phenomena are not tied to measurement alone and are actual. The discrepancy in dark counts is due to a difference in spectral range between the devices. In the original manuscript, we mentioned that the 89C device extended further into the UV range of the EM spectrum; however, we failed to clearly link this fact with the resulting higher photon counts due to the wider sampling band. We agree that this is a relevant detail and have included explanation under the 2.3 Photomultiplier Measurements heading. To maximize the clarity of our reported data, we used the untransformed raw values for both PMTs.

2) Comment: The variation from the PMTs (low dark counts, low QE, UV-VIS) or (high dark counts, high QE, VIS) in Tables 1 and 2 is not entirely intuitive. Table 1 and Table 2 are the same except for the last four rows!
2) Response: The reviewer is correct. This was an error that occurred when configuring the tables. The revised manuscript contains the appropriate tables with accurate data. In addition, standard deviation values have been added to the tables.

3) Comment: Labels in Figures 2 and 3 are not consistent. Use"cancer" or "malignancy".
3) Response: We agree that the nomenclature should be consistent across graphs and the manuscript as a whole. We have therefore replaced instances of “(non-)malignant” with “(non-)cancerous” in the revised manuscript.

4) Comment: The x-axis for Figure3 is missing.
4) Response: Figure 3 has been re-arranged in the revised manuscript and the x-axis labels are now included.

5) Comment: Figure 3 bar plot doesn't give any additional information than figure 2, without details on the density estimation.
5) Response: As stated under comment #4, we have re-arranged the figure in the revised manuscript.

6) Comment: Fig5: Consider using color-coding or similar to help read the graph easier.
6) Response: The suggestion is well-received. We have changed the figure to include colour-coding to increase at-a-glance readability.

7) Comment: 50Hz sampling: is this limited by the amplifier or the USB bandwidth? This seems too low sampling for most biomedical imaging units (1-200 kHz). Was this PMT unit a cost-efficient implementation?
7) Response: While the PMTs are capable of measuring in the MHz range, the upper limit of the software was 50Hz. The revised manuscript now includes the name of the software (Page 4) The approach was meant to be cost-efficient and accessible as, from a biomedical perspective, increasing access to the technology is an important feature. That significant findings were evident with such a low temporal resolution is, in our view, further evidence of the robustness of the phenomenon since we would expect higher resolution measurements to reveal narrower bands of spectral power with finer discriminating potential.

8) Comment: How was the discriminant analysis performed? No details on the method are presented. Why are the spectral graphs/power density graphs not presented?
8) Response: The discriminant analyses performed were linear step-wise discriminant analyses (this has been inserted into the methods section). The graphs were no presented as we were only comparing 2 or 3 groups, and felt the statements “[18(6.7) vs. 23(6.7) vs. 25.7(7.2) respectively; Means and SD; p=0.033, eta2=.15]’ accurately depicted the relationship. The results, thematically, were also heavily consistent with the previously reported data. For this reason, we again thought it would be prudent to limit the number of graphs present while maintaining the efficacy of presenting the significant data.

9) Comment: If authors are willing, I suggest pushing the time-series data to open-analysis/ open-data platforms for community efforts. Also, additional work to identify the significance of these count rate differences, apart from a regular bar plot, would add some rigor to the analysis.
9) Response: This is an excellent suggestion. We are in the process of compiling a larger aggregate dataset for community engagement. Once available, we expect the expertise of researchers in fields of bioinformatics and related disciplines will allow for a more thorough and multifaceted examination of the data. Increased sampling rates may be needed in future applications.

10) Comment: Line19: "photons per" -> it is not clear what authors mean here (area, time or others)
10) Response: The word “per” has been removed in the revised manuscript to make the sentence coherent.

Reviewer 3 Report

This is a good paper on distinction of cancer cells from non cancerous ones. I recommend publication of this article with implementation of 1 major (without addressing this major concern the statistical analysis is technically flawed and should not be published) and 2 minor edits:

Major edit:

  1. Why authors do not show standard deviation (SD) and why is  standard error of mean (SEM) preferred for comparison ? Especially if in figure 5 SD is presented instead of SEM I doubt if controls and tumour results would be comparable statistically with AVONA test. 

SD measures the amount of dispersion for a subject set of data from the mean, while the SEM measures how far the sample mean of the data is likely to be from the true population mean.  So unless you can justify that trueness of a data is more important than the actual variability in your measurement analysis, the choice of SEM is not correct. 

Minor edits:

  1. Significance analysis tests, such as the AVOVA, T-test, should be applied to the tables and to the comparison of bars in the figure 2 & 3.
  2. What is the physical importance of discriminant score ? Discuss.

Author Response

We have provided a point by point response to your comments. We thank you for your time and helpful comments, the paper is truly in a better place because of your suggestions. 

Reviewer #3
We would like to thank the reviewer for their comments and suggestions regarding the statistics. The paper has significantly improved thanks to your suggestions. The revised manuscript includes edits that address their comments.
Major Edit
1) Comment: Why authors do not show standard deviation (SD) and why is standard error of mean (SEM) preferred for comparison ? Especially if in figure 5 SD is presented instead of SEM I doubt if controls and tumour results would be comparable statistically with AVONA test.
1) Response: This is a sensible distinction and we agree that SDs are highly relevant as the intrinsic variability of the sample indicates the degree to which values are reliable. We have therefore included SD values throughout the revised manuscript text (e.g., Tables 1 and 2; final paragraph of results section on Page 8). When graphing our results, however, we decided to use SEMs to reflect the statistical tests we performed. While SDs were in many cases quite high, in our previous experience this is not due to inter-trial variability but rather an intrinsic feature of measuring UPEs from biological sources with the PMTs. Because the measurement is relatively crude (though clearly not so crude as to obscure the effects), moderate-to-high variability is expected and therefore SDs within samples can be high. The SEMs presented in each graph, as you correctly point out, indicate the approximation to the population mean – which is, in our view, a more important metric in this context. With SEMs, the statistical tests can be graphed and real differences between the conditions can be visualized easily. With SDs, visualization of real differences between the groups is more difficult despite real differences between the groups.

Minor Edits
1) Comment: Significance analysis tests, such as the AVOVA, T-test, should be applied to the tables and to the comparison of bars in the figure 2 & 3.
1) Response: Tables 1 and 2 have been reformatted in the revised manuscript and now include SDs; we did not include statistical statements to avoid over-crowding of the table. In the revised manuscript, mentions of Figures 2 and 3 now include statistical statements.

2) Comment: What is the physical importance of discriminant score ? Discuss.
2) Response: In the context of Figure 3B, the scores visually demonstrate the degree to which the samples differ as well as their within-group variabilities. It was added as an additional visual representation to demonstrate the dispersion, and lack of carryover between scores for the cancerous vs non-cancerous conditions.

Round 2

Reviewer 2 Report

The manuscript has revised its presentation and the authors have answered all my concerns satisfactorily. I have no further review comments. I find this work exciting to the biomedical engineering community. Good luck and I appreciate the authors' efforts in the review process and I wish the best for their continuing research in this field.

Reviewer 3 Report

All concerns addressed. The paper can now be published.